# Development and Evaluation of the “High-Up” Program for Enhancing the Nursing-Management Competency of Mid-Career Hospital Nurses: A Quasi-Experimental Study

**DOI:** 10.3390/ijerph19074392

**Published:** 2022-04-06

**Authors:** Seulki Kim, Ji-Young Lim

**Affiliations:** 1Department of Nursing, Seoil University, 28, Yongmasan-ro 90-gil, Jungnang-gu, Seoul 02192, Korea; seulki920111@naver.com; 2Department of Nursing, Inha University, 100, Inha-ro, Michuhol-gu, Incheon 22212, Korea

**Keywords:** nurse, nursing management, problem-based learning, competency-based education, internet-based intervention

## Abstract

The aim of this study was to develop an educational program to strengthen the nursing management competency of experienced nurses who are prospective nurse managers and then determine the effectiveness of the program. This quasi-experimental study was conducted from January to April 2021. A total of 22 nurses were assigned to the experiment group (mean age: 26.55 ± 1.30 years; 2 males, 20 females), and 20 were assigned to the control group (mean age: 27.55 ± 2.04 years; 20 females). The program, known as the “High-Up” program, comprised problem-based learning (PBL) and video lectures. In the experiment group, nurses discussed PBL cases through video conferences and applied problem-solving methods. The collected data were analyzed using the Friedman test and Wilcoxon rank-sum test (administered through SPSS). At four weeks after the intervention, the experiment group showed higher critical thinking tendency scores than the control group (pre-intervention score: 3.48 ± 0.36; post-intervention score: 3.71 ± 0.49; Z = −1.99, *p* = 0.046). The findings indicate that the “High-Up” program can enhance the nurse management competency of experienced nurses who need to prepare for nurse manager roles, and that it can also positively influence the performance of nursing organizations. However, it can be difficult to comprehensively enhance nursing management competency in a short period of time, meaning continuous education is required.

## 1. Introduction

Between 2008 and 2018, the number of nurses in South Korea increased by approximately 150,000 (246,840 to 394,662) [1]. Accordingly, the nurse manager role, which involves managing and supervising the nursing workforce, is becoming increasingly important, and as the nursing workforce continues to grow, such managers urgently require systematic human resource management capabilities [2]. However, prospective nursing managers must be appropriately trained for their new roles. This is because the transition from a practice nurse to a nurse manager is associated with fear and stress [3], and because the management competency of nurse managers can affect the turnover intention and motivation of the nurses they supervise, which can consequently impact the quality of nursing care and productivity at health-care organizations [4].

Systematic education and training programs for nursing managers are important to strengthen nursing managers’ management competency and, consequently, improve nurses’ job performance and benefit nursing organizations [5]. Korean hospitals provide education and training programs for all new nurse managers; however, the programs used are generally self-developed and usually insufficient to provide managers with sufficient skills to effectively manage growing nursing organizations [6].

In contrast, in the US, the Fundamental Skills for Nurse Managers (FSNM) course has been developed to help foster competent nurse managers [7]. This is a fully online course that aims to strengthen new managers’ theoretical knowledge and practical skills and has been successful in its pursuit. Nurse manager training programs, such as the FSNM, which provide guidelines for effectively nurturing competent nurse managers within a short time period, can serve as models for the development of similar programs for nurses in other countries.

To strengthen prospective nurse managers’ ability to solve novel, complex organizational management problems, they should be exposed to various teaching and learning methods centered on actual cases. Problem-based learning (PBL) is a self-directed learning strategy that provides learners with enhanced responsibility and autonomy and asks them to solve problems through small group cooperation [8].

Blended learning concerns combining two or more teaching techniques or delivery methods to enhance learning content and learners’ learning experiences [9]. Such learning can comprise both online and face-to-face (i.e., offline) learning methods [10]. The increased demand for non-face-to-face education as a result of measures to control the coronavirus disease 2019 (COVID-19) pandemic has led to an expansion of online education; the need for remote education is especially strong for hospital nurses, as this group has a particular need to limit external contacts. Consequently, e-PBL, which comprises PBL in a web-based environment, is becoming increasingly popular [11,12].

Several previous studies have focused on creating educational programs for nurse managers. For instance, Abraham [13] developed a six-month program that taught nurses theoretical knowledge, core competencies, and leadership skills, with the aim of improving the nurses’ leadership and professional behaviors. Nine months after the program’s completion, 20% of the participants had been promoted to nursing managers within the target organization, a much higher percentage than the 0.3% among the organization’s general staff nurse population [13]. Additionally, Frasier [14] tested a pilot program for building leadership competency among early nurse managers and found that the program increased the participants’ self-awareness scores (from 12.36 to 13.24 points). However, many studies in this field have not specifically focused on nursing management capabilities but on leadership, indicating the lack of data on programs for strengthening the unique characteristics of nursing management capability.

The present research aimed to develop an educational program for strengthening the nursing management competency of experienced nurses who are prospective nurse managers and to evaluate the program’s effectiveness, particularly in regard to communication ability, critical thinking tendency, and problem-solving ability. This program, named “High-Up,” integrates nursing management core competencies (identified through a systematic literature review), uses the FSNM as a theoretical framework, and features blended e-PBL.

### 1.1. Fundamental Skills for Nurse Managers

The FSNM is an educational program that provides guidelines for nurse managers. It was developed by the American Organization for Nursing Leadership and focuses on mid-careerists. The FSNM provides learners with the confidence and knowledge to grow from general nurses to nurse managers by emphasizing five core competencies: getting started as a new nurse manager, human resource management, financial management, quality and safety, and leadership [7]. The FSNM comprises an online modular design that allows adjustments depending on learners’ needs and minimizes temporal and spatial requirements.

Shen et al. [15] used the FSNM framework to create a nursing leader residency program that focused on nursing skills, science, and individual leadership characteristics; they consequently found that interpersonal relationship management and communication with other nurses are important leadership development focuses. Meanwhile, a study that applied the Stanford Health Care Leadership Development Workshop to charge nurses found that, at the six-month follow-up, the participants’ leadership, decision-making ability, and scientific knowledge had improved, as did their competency as educators and practitioners [16].

The FSNM characteristics and aims accord with the purpose of this study; therefore, the FSNM was used as a theoretical framework for the “High-Up” program.

### 1.2. Systematic Review

To identify and confirm the nursing management competency concepts that would form the basis of the “High-Up“ program, a systematic literature review was conducted, focusing on academic literature published in the last 10 years, from January 2010 to April 2020. The searched databases included Ovid-EMBASE and Ovid-MEDLINE, as well as South Korea-based academic databases, such as RISS, NDSL, and KISS. The search was performed by combining keywords such as “nurses”, “management competency” (“capability”, “ability”, “capacity”), “program”, “empowerment” (“reinforcement”, “enhancement”), “development”, and “advancement” (“improvement”).

The population, intervention, comparator, and outcomes (commonly known as “PICO”) approach was used for literature extraction; the population was nurses, the intervention was a nursing management competency program, and the comparator and outcomes were not limited. The PICO strategy can be used to construct several types of research questions originating from clinical practice [17]. The inclusion criteria were studies (1) targeting nurses, (2) focused on nursing management competency programs, and (3) involving quantitative research. The exclusion criteria were studies that (1) were not published in Korean or English, (2) were not original articles, (3) featured animal experiments, and (4) were preclinical studies.

The search returned 38,535 articles, including 28,489 from Medline, 13,559 from Embase, 141 from RISS, 8 from NDSL, and 2 from KISS. First, 34,882 duplicate articles were excluded. Second, after removing duplicate articles, 3653 full texts and abstracts were confirmed. Third, 3344 articles deemed not to be related to nursing management competency programs for nurses were excluded. The inter-rater agreement for this decision was 0.92 (kappa coefficient). Fourth, from the remaining 309 documents, 284 were excluded. The reasons for the exclusion were as follows: 2 studies not targeting hospital nurses, 7 studies where the original text could not be obtained, 257 studies that did not apply intervention, and 18 studies that were not appropriate to be included in the content analysis. Twenty-five articles were left for extraction. The agreement between evaluators at this stage was 0.91. Finally, the remaining 25 articles were evaluated for quality using the methodological checklist version 2.0 of the Scottish Intergroup Guidance Network (SIGN) [18]. The 25 articles were divided into 2 articles evaluated as (++), 10 articles evaluated as (+), 11 articles evaluated as (−), and 3 articles that were rejected. Therefore, a total of 12 articles were included in the final analysis.

For the final extracted literature sample, content analysis was conducted on participant numbers, program contents, major measurement variables, program operation methods and time periods, pre- and post-program scores, and evaluation methods. The core elements of the “High-Up” program were selected by combining the results of this analysis with the FSNM framework. The overall research process for this study is summarized in Figure 1.

### 1.3. Hypotheses

The following hypotheses were set for this study:

**Hypothesis 1:** The experiment group, which is the group that participates in the “High-Up” program, will show significantly higher communication ability scores at the first post-test (T1), the second post-test (T2), and the third post-test (T3) when compared to the control group, which does not participate in the program.

**Hypothesis 2:** The experiment group will show significantly higher critical thinking tendency scores at T1, T2, and T3 when compared to the control group.

**Hypothesis 3:** The experiment group will show significantly higher problem-solving ability scores at T1, T2, and T3 when compared to the control group.

## 2. Materials and Methods

### 2.1. Design

This was a quasi-experimental study with a repeated-measures design, a non-equivalent control group, and pre- and post-tests. The aim was to develop a nursing management competency-strengthening program for nurse managers named “High-Up” and to analyze its effectiveness after administering the program for five weeks to experienced nurses working in hospitals.

### 2.2. Sample

The participants were experienced nurses working in small- and medium-sized hospitals with less than 500 beds, where systematic education programs for career nurses were scarce. The sample size was calculated using G*Power 3.1.9.2. To afford repeated-measures analysis of variance (ANOVA) of an experiment and control group, the power was set to 0.90, the significance level to 0.05, the effect size to 0.5, the number of groups to two, and the number of repeated measurements to four. Additionally, to afford within–between interaction analysis and difference analysis between the two groups, a calculation was performed with a power of 0.80, a significance level of 0.05, and an effect size of 0.5 [19]. To estimate the number of participants required to achieve statistical significance in the study, G*power 3.1.9.2 was used [20]. The calculation for the repeated-measures ANOVA returned a minimum sample size of 10 per group, for a total of 20 participants, while the calculation for the difference analysis returned a minimum sample size of 18 per group, for a total of 36 participants. The target number of participants was consequently determined to be 44, allowing for a dropout rate of approximately 20%.

The inclusion criteria for participation were nurses who (1) worked in small- or medium-sized hospitals or general hospitals with 101–499 beds; (2) had 3–5 years of experience (this range was chosen because, according to Benner’s [21] expert role theory, nurses with such experience have gained the ability to view issues from a comprehensive perspective and have reached the stage of competence); (3) were career nurses, not nursing managers; and (4) understood the purpose of this study and agreed to participate. The participants were recruited through online announcements made on nursing department social network systems and offline announcements posted on the bulletin boards of the nursing department in three hospitals. The specific research process was explained to individuals who expressed an interest in participating. If an individual wanted to participate in the “High-Up” program to strengthen their nursing management competency, they were assigned to the experiment group. If they wished to participate in some of the research processes but not the program itself, they were assigned to the control group.

After recruitment, three members of the experiment group left the group for personal reasons, and one withdrew their intention to participate for an undisclosed reason. In the control group, three participants withdrew their intention to participate, while three did not respond during the collection of the questionnaires for the third post-test. Therefore, their data were excluded from the analysis. Thus, 22 members of the experiment group completed the “High-Up“ program and all surveys, and 20 members of the control group completed the control group material and all surveys.

### 2.3. Intervention

#### “High-Up” Program Development

The leadership model of the Healthcare Leadership Alliance (HLA), a consortium of major professional associations in the fields of nursing and medicine, was used to select content topics for the “High-Up” program. Specifically, the five-session program was constructed based on the sub-domains of the HLA model: communication and relationships, knowledge of the health-care environment, leadership, business skills and principles, and professionalism.

The “High-Up” program was online-based and provided PBL through video conferences and lectures. The video conferences lasted 120 min each week, and concerned PBL cases were selected to represent each targeted topic. Following the PBL procedure, the program proceeded in the order of team building, presenting problems, identifying problems, writing a study plan, self-directed learning and team activities, completing and reviewing solutions, and presenting, evaluating, and reflecting on solutions [22].

The PBL cases used in the lectures were examples of practical cases that commonly occur in clinical settings; these were chosen to strengthen the nurses’ nursing management competency. Additionally, for effective discussion in the e-PBL classes, participants were assigned the roles of moderator, logicist, idea person, intuitionist, optimist, and pessimist throughout the entire discussion, which encouraged the learners to be actively engaged. Six colored hats, one of the creative thinking techniques, were used [23]. The roles before the start of e-PBL were divided so that neutral, negative, emotional, optimistic, creative, and rational thinking could be carried out, respectively. At the end of each week, learners were asked to write a reflection log, which allowed them to reflect on their experiences and things they had learned, and the instructor provided feedback on these logs. Table 1 summarizes the overall outline of the learning process.

### 2.4. Content Validity Evaluation

The construct and content validities of the “High-Up” program were evaluated by a panel of 15 experts, which included nursing professors and nurse managers. Content validity was evaluated using the content validity index (CVI) [24], with one point awarded for “very not valid,” two for “not valid, “three for “valid, “and four for “very valid.” All items received CVIs of ≥0.80, indicating reasonable program content. The items with the highest validity, at 0.96, were patient safety and evidence-based practice in the “knowledge of the health-care environment” domain. The item with the lowest validity, at 0.83, was influential behavior in the “communication and relationship management” domain.

### 2.5. Application

The “High-Up” program was applied over five weeks, from January 18 to 21 February 2021. Educational materials and online training videos were provided to the experiment group at the beginning of the program, and each week, they were asked to participate in the real-time online learning (e-PBL) after performing advanced self-learning of the content to be discussed that week. The e-PBL was conducted for two hours each week, with the same content being provided on three different days (Monday, Wednesday, and Thursday); this allowed the participants to choose a day that suited their work schedule, thereby facilitating participation. Additionally, to encourage self-directed learning, participants were asked to log the completion of each video lesson on social media, with these learning progress reports being checked regularly. Similar to the experiment group, the control group was provided with the “High-Up” program textbook, which allowed them to engage in self-directed learning over five weeks; only the experiment group received the intervention.

### 2.6. Measurement

#### 2.6.1. Communication Ability

Communication ability was measured using a scale developed by Lee et al. [25]. This scale comprises 49 items, each of which is scored using a five-point Likert scale; higher total scores indicate higher communication ability. Cronbach’s α for the scale was 0.80 at the time of development and 0.87 for this study.

#### 2.6.2. Critical Thinking Tendency

The participants’ critical thinking tendency was measured using a scale developed by Yoon [24]. This scale comprises 27 items, each of which is scored using a five-point Likert scale; higher total scores indicate higher critical thinking tendency. Cronbach’s α for the scale was 0.85 in Yoon’s study [26] and 0.84 in this study.

#### 2.6.3. Problem-Solving Ability

Problem-solving ability was measured using a scale developed by Lee et al. [25]. This scale comprises 45 items, each of which is scored using a five-point Likert scale; higher total scores indicate higher problem-solving ability. Cronbach’s α for the scale was 0.94 at the time of development and 0.89 for this study.

### 2.7. Data Collection

Data were collected from 11 January to 18 April 2021. The questionnaire used for data collection was administered at the pre-test, which was held two weeks prior to the start of the program (T0); at the first post-test, which was held on the same day as the completion of the program (T1); at the second post-test, which was held four weeks after the completion of the program (T2); and at the third post-test, which was held eight weeks after the completion of the program (T3).

### 2.8. Data Analysis

Data analysis was performed using IBM SPSS/WIN 26.0. Descriptive statistics were analyzed using frequencies, percentages, means, and standard deviations. Wilcoxon’s rank-sum test and Fisher’s exact test were used to test the homogeneity of the variables. The Kolmogorov–Smirnov test was used to test normality; this indicated that the communication ability, critical thinking tendency, and problem-solving ability variables did not follow normality. A non-parametric analysis was performed on these variables. The Friedman test was conducted on each group to verify the effectiveness of the program. Wilcoxon’s rank-sum test was used to examine the degree of change in the post-test values when compared to the pre-test. A two-sided cut-off of 0.05 was used for the *p*-value.

### 2.9. Ethical Considerations

This study was approved by the institutional review board of Inha University, Incheon, South Korea (200218-1AR). After obtaining the prior consent of the hospital’s nursing department, the recruitment notice was uploaded to the nursing department social network systems or ward bulletin boards. Nurses who wished to participate were given an explanation of the purpose of the study, the data collection method, and their right to choose not to participate and to withdraw consent at any time during the study. Only individuals who provided written consent to participate were included in the sample. To remove the risk of influence from the authors on the program’s intervention effect, a separate trained research assistant was in charge of administering the curriculum.

## 3. Results

### 3.1. General Characteristics and Homogeneity Test

Table 2 presents the participants’ general characteristics. The sample was homogeneous regarding communication ability, critical thinking tendency, and problem-solving ability (Table 2).

### 3.2. Hypothesis Tests

Hypothesis 2 was partially supported, as there was a statistically significant intergroup difference in critical thinking tendency at some time points ($\chi^{2}$ = 11.41, *p* = 0.010); specifically, a statistically significant intergroup difference was identified when comparing T1 and T0 scores for critical thinking tendency, partially supporting Hypothesis 2 (Z = −1.99, *p* = 0.046). In contrast, the intergroup differences associated with Hypothesis 1 and Hypothesis 3 were not statistically significant (Table 3, Figure 2).

## 4. Discussion

When developing the “High-Up” program, the results of a systematic literature review and the core concepts of the FSNM framework were integrated to set goals and organize the contents. The program was developed to include the educational elements necessary to strengthen nurses’ nursing management competency, and its content validity was verified by a panel of experts. Applying e-PBL, a non-face-to-face teaching/learning method, the program included learning activities for nurses and was based on small group interactions, which can improve individuals’ ability to collaborate with various expert teams and encourage active and voluntary participation [27].

After the completion of the “High-Up” program, three post-tests were conducted to investigate changes in variables such as communication ability, critical thinking tendency, and problem-solving ability. These variables were targeted based on the results of a study that showed a significant positive relationship between empirical learning and nursing management competency [28]. Our analysis showed that the “High-Up” program supported some of our hypotheses, having a statistically significant effect on critical thinking tendencies. Below, we conduct an in-depth discussion of these results and present implications for future research.

Analysis of the effects of the “High-Up” program revealed that at the first post-test, there was a significant intergroup difference in critical thinking tendency. In the “High-Up” program, the participants analyzed PBL cases and then applied a problem-solving process based on objective evidence. The observed improvement in critical thinking ability appears to be a result of the participants’ training in identifying the causes of problems and determining priority solutions. This accords with the results of a previous study on nursing graduate students, which reported that the use of a free learning process improved the participants’ creative thinking and diversity of ideas [29].

In the rapidly changing hospital environment, nurse managers play an important role in managing nursing resources and systems [30]. Critical thinking ability has an important influence on decision making and creative problem-solving ability in clinical settings [31]. Notably, the Joint Commission International accreditation for the field of nursing sets critical thinking as a core skill, which shows that it is an important management competency that must be continuously explored and developed in clinical settings [32,33]. Thus, an educational approach for cultivating nurse managers’ creative and critical thinking abilities is essential for such managers to effectively negotiate the various problems and conflicts that occur in clinical settings [34]. PBL is one such learning method, as it exposes learners to similar situations to those they will encounter in real life [35].

In this study, the communication ability score immediately after the intervention (T1) was 3.80 ± 0.37 in the experiment group and 3.59 ± 0.29 in the control group, showing no statistically significant intergroup difference. However, when compared to the pre-test, the control group’s score had increased by 0.07 ± 0.06, while the experiment group’s score had increased by much more, 0.14 ± 0.12. This result is partially consistent with that of Jang et al. [35], who argued that team-based learning has a positive effect on improving communication ability. Communication can facilitate rational decision making and the establishment of therapeutic relationships; while both are vital, the latter is an important factor for the psychological stability of individuals who form mutual relationships within large-scale organizations and can affect not only job performance, but also ability and morale enhancement [36,37].

Regarding problem-solving ability, both groups showed changes in average scores across the time points. Both groups showed an increase in scores at the third post-test. In particular, among the seven subareas of critical thinking tendency, including intellectual passion/curiosity, prudence, confidence, systematicity, intellectual fairness, sound skepticism, and objectivity, there was a significant difference between the groups in terms of intellectual fairness [38]. The control group also showed an increase in its post-test scores when compared to its pre-test scores. The control group was provided with textbooks describing the learning goals, lecture outlines for each week, and educational content. Thus, the control group’s score seems to have increased as a result of the theory-based learning.

Regarding limitations to this study, it is regrettable that the hypotheses were only supported with regard to critical thinking tendency. Nursing management competencies include abstract and complex concepts that are difficult to change in a short period of time. Therefore, the provision of a one-time, five-week intervention may have been insufficient to significantly impact communication ability and problem-solving ability. Thus, a future study that, after applying the program, features subsequent booster classes is required. This study was designed as a non-equivalent control group, and pre- and post-tests and random assignments were not carried out. Therefore, there will be limitations in the generalization of the research results.

## 5. Conclusions

The “High-Up” program developed in this study showed a positive effect on the critical thinking tendency of prospective nurse managers. Furthermore, the fact that the content of the evidence-based program was constructed by combining the results of a systematic literature review and the FSNM framework indicates it can represent useful material for the development of follow-up education programs for strengthening nursing management capabilities. In terms of methodology, e-PBL, which combines self-learning, video education, and real-time online lectures, was applied to accommodate the need for online teaching and learning environments during the COVID-19 pandemic. In particular, e-PBL is a flexible educational resource that can be used regardless of time and place, overcoming the limitations of face-to-face education. This can be especially important in the nurse education context, as the nursing profession often involves three-shift rotation schedules, meaning nurses can have difficulty attending formally scheduled classes. Nursing management competency is a complex concept that is difficult to change in a short period of time. Therefore, the development of an education and training system that can provide continuous, boosting education is required. The “High-Up” program has been developed for experienced nurses who are prospective nurse managers; it will be necessary in the future to develop in-depth programs tailored to various other types of nurses with diverse clinical experience and prior knowledge.

## Figures and Tables

**Figure 1 ijerph-19-04392-f001:**
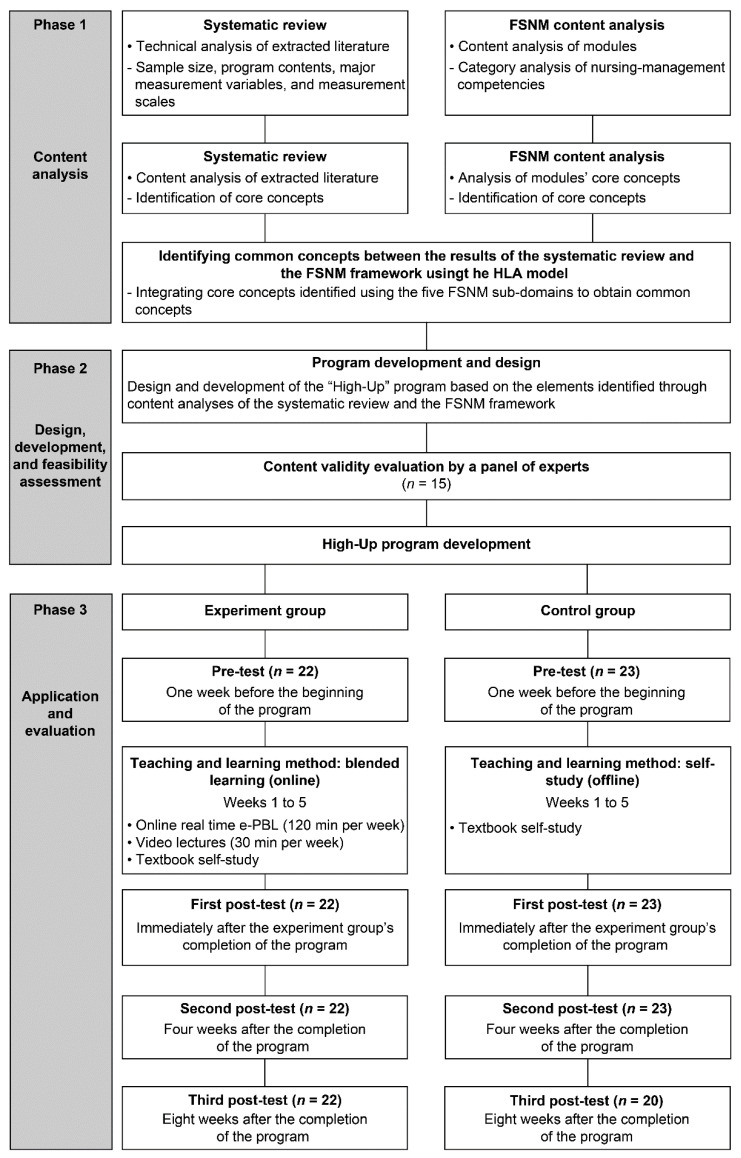
Chart outlining the development process for the “High-Up” program and the testing of its effects. FSNM: Fundamental Skills for Nurse Managers; HLA: Healthcare Leadership Alliance; PBL: problem-based learning.

**Figure 2 ijerph-19-04392-f002:**
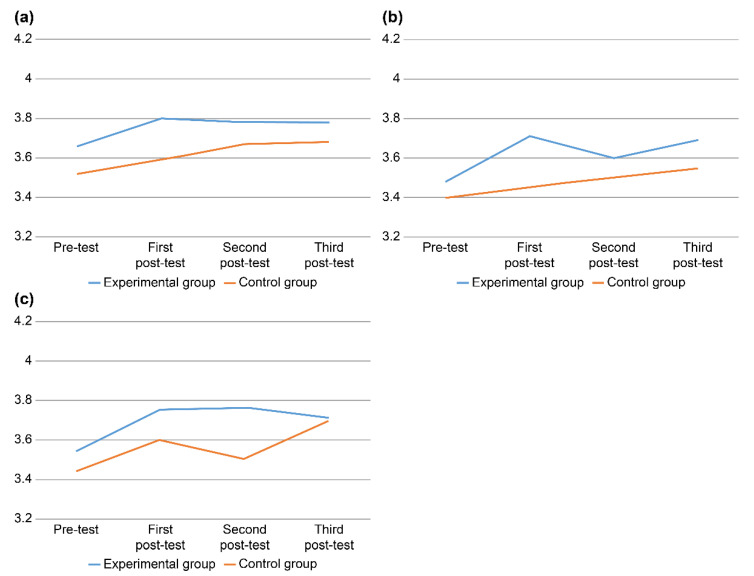
Effects of the “High-Up” program on the (**a**) Communication ability, (**b**) Critical-thinking tendency, (**c**) Problem-solving ability.

**Table 1 ijerph-19-04392-t001:** Overview of the program process by week.

Week	Main Theme	Contents	Time (min)	Method
1	Communication and relationship management	Building trust relationships Understanding the difference between generations and occupations Effective communication Importance of exercising influence	10 5 10 5	Video lectures
		Guide to basic learning, PBL Team building Presenting a problem Identifying the problem Planning study Self-directed learning and team activities Finding and reviewing solutions Presentation Evaluation and reflection	5 15 10 10 10 20 20 20 10	Online real-timee-PBL
2	Knowledge of the health-care environment	Patient safety accidents Service quality management and PDCA Systematic literature review and evidence-based practice Nursing in the era of the 4th Industrial Revolution Older-adult population growth and policies	5 5 5 10 5	Video lectures
		Icebreaking (similar to the catch mind game) Presenting a problem Identifying the problem Planning study Self-directed learning and team activities Finding and reviewing a solution Presentation Evaluation and reflection	15 10 10 15 20 20 20 10	Online real-timee-PBL
3	Leadership	Critical thinking Efficient decision making Establishment of organizational structures and strategies Leadership	5 10 5 10	Video lectures
		Icebreaking (character quiz) Presenting a problem Identifying the problem Planning study Self-directed learning and team activities Finding and reviewing a solution Presentation Evaluation and reflection	15 10 10 15 20 20 20 10	Online real-timee-PBL
4	Business skills and principles	Financial management Human and material resource management Privacy protection	10 10 10	Video lectures
		Icebreaking (cry in silence; similar to lip reading) Presenting a problem Identifying the problem Planning study Self-directed learning and team activities Finding and reviewing a solution Presentation Evaluation and reflection	15 10 10 15 20 20 20 10	Online real-timee-PBL
5	Professionalism	Overview of nursing ethics Professionalism	20 10	Video lectures
		Icebreaking (You too? Me too!) Presenting a problem Identifying the problem Planning study Self-directed learning and team activities Finding and reviewing a solution Presentation Evaluation and reflection	10 10 10 10 15 15 15 10 25	Online real-timee-PBL

PDCA: plan, do, check, act; Icebreaking (You too? Me too!): The participants were given a subject word and had to write 15 words related to it. The participants were then required to sing one word at a time, and if the word was identical to the words sung by other participants, they had to delete that word from the list. The first person to erase all the words would win.

**Table 2 ijerph-19-04392-t002:** Participants’ general characteristics and homogeneity test.

Variable		Exp (*n* = 22)	Con (*n* = 20)	$Z/\chi^{2}$	*p*
		*n* (%) M ± SD	*n* (%) M ± SD		
Gender	Male	2 (9.1)	0 (0.0)	1.91	0.489
	Female	20 (90.9)	20 (100.0)		
Age		26.55 ± 1.30	27.55 ± 2.04	−1.85	0.065
Religion	Yes	6 (28.6)	1 (5.3)	3.75	0.095
	No	15 (71.4)	19 (94.7)		
Married	Yes	0 (0.0)	1 (5.0)	1.13	0.476
	No	22 (100.0)	19 (95.0)		
Education	College	1 (4.5)	5 (25.0)	3.58	0.087
	University	21 (95.5)	15 (75.0)		
Clinical career	<4 years	18 (81.8)	11 (55.0)	3.83	0.300
	≥4 years	4 (18.2)	9 (45.0)		
Communication ability		3.66 ± 0.05	3.52 ± 0.08	−1.66	0.096
Critical thinking tendency		3.48 ± 0.08	3.40 ± 0.07	−1.15	0.251
Problem-solving ability		3.54 ± 0.07	3.44 ± 0.07	−0.87	0.385

Con: control group; Exp: experiment group.

**Table 3 ijerph-19-04392-t003:** Summary of the test results for intergroup differences.

Variable	Group	T0	T1	T2		T3	Friedman
		M ± SD	M ± SD	M ± SD		M ± SD	$\chi^{2}(p)$
Communication ability	Exp	3.66 ± 0.25	3.80 ± 0.37	3.78 ± 0.39		3.78 ± 0.38	3.22(0.359)
	Con	3.52 ± 0.35	3.59 ± 0.29	3.67 ± 0.28		3.68 ± 0.30	2.22(0.527)
	Difference Z(*p*)		T1–T0	T2–T0	T3–T0		
			−1.00(0.319)	−0.04(0.970)	−0.37(0.715)		
Critical thinking tendency	Exp	3.48 ± 0.36	3.71 ± 0.49	3.60 ± 0.54		3.69 ± 0.47	11.41(0.010)
	Con	3.40 ± 0.32	3.46 ± 0.29	3.50 ± 0.36		3.55 ± 0.37	1.17(0.760)
	Difference Z(*p*)		T1–T0	T2–T0	T3–T0		
			−1.99(0.046)	−0.35(0.724)	−0.56(0.579)		
Problem-solving ability	Exp	3.54 ± 0.31	3.75 ± 0.41	3.76 ± 0.47		3.71 ± 0.44	8.52(0.036)
	Con	3.44 ± 0.32	3.60 ± 0.39	3.50 ± 0.35		3.69 ± 0.35	12.26(0.007)
	Difference Z(*p*)		T1–T0	T2–T0	T3–T0		
			−1.60(0.110)	−0.82(0.413)	−0.72(0.473)		

Con: control group; Exp: experiment group.

## Data Availability

The data presented in this study are available on request from the corresponding author. The data are not publicly available due to privacy and ethical restrictions.

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
