# Peer review of "Development and Evaluation of the “High-Up” Program for Enhancing the Nursing-Management Competency of Mid-Career Hospital Nurses: A Quasi-Experimental Study"

_ijerph, 2022, doi:10.3390/ijerph19074392_

Round 1

Reviewer 1 Report

The objective of this study was to develop an educational program to strengthen the nursing management competence of experienced nurses who are future nursing managers and then determine the program's effectiveness. It shows the implementation of an educational program to solve the deficit of nursing managers.
It is a scientifically sound manuscript with an appropriate experimental design to test the hypothesis.
The conclusions are consistent with the evidence and arguments presented.
Therefore, I recommend publishing the article after checking /reviewing some minor details marked in yellow in the attached pdf document.

Author Response

March 26, 2022

Dear reviewer

We wish to re-submit the manuscript titled “Development and Evaluation of the “High-Up” Program for Enhancing the Nursing-Management Competency of Mid-Career Hospital Nurses.” The manuscript ID is ijerph-1633005.

We thank you and the reviewers for your thoughtful suggestions and insights. The manuscript has benefited from these insightful suggestions. I look forward to working with you and the reviewers to move this manuscript closer to publication in the International Journal of Environmental Research and Public Health.

The manuscript has been rechecked and the necessary changes have been made in accordance with the reviewers’ suggestions. The responses to all comments have been prepared and attached in the response letter.

Thank you for your consideration. I look forward to hearing from you.

Sincerely,

Ji Young Lim

Department of Nursing, Inha University

100 Inha-ro, Michuhol-gu, Incheon, 22212, Republic of Korea

Tel: +82-32-860-8210

Email: lim20712@inha.ac.kr

Reviewer 2 Report

I want to congratulate the team of authors for the study carried out on a relevant and, at the same time, necessary topic, such as the training and development of effective leadership skills in managerial professionals, specifically in nursing leaders.

It represents a further step in the study and professional updating and, on the other hand, the development of new tools based on problem-based learning adds value to the improvement of leadership skills.

After reviewing the article, I write down my suggestions and recommendations:

  • The summary must be adjusted to a maximum of 200 words.
  • It is recommended to structure the work in Introduction, Materials and Method, Results, Discussion and Conclusions. In this sense, it is suggested to move the hypotheses and the general objective to the end of the Introduction, before the Materials and Methods section.
  • Before the hypotheses, a paragraph appears (lines 84-88), which refers to the findings of the study, that paragraph should go to the Results or Conclusions section, one should not talk about findings when the phases have not yet been developed. methodological and empirical research.
  • In line 137, data written numerically and textually are combined, the way of annotating the data numerically must be unified.

It would be convenient to continue delving into the subject with experimental studies that specifically relate problem-based learning and the development of competency in nursing leadership.

A cordial greeting.

Author Response

(The authors gave the same response as above.)

Reviewer 3 Report

The research is very interesting and well-designed, deals with very important and current topics, and deserves publication.

I add some comments and suggestions for authors.

Abstract:
Be consistent in numerically reporting results at averages and standard deviations.

Introduction:
Add more up-to-date literature (mostly from the last 5 years).
In Hypothesis 1, you do not need to use the terms experimental and control group twice.

Theoretical Framework:

2.2. Systematic Review
Add a citation for the PICO question.
I recommend that you add an inclusion criterion related to the type of research (for example, quantitative research) instead of the inclusion criterion 3) presenting statistical results.
You wrote that 284 articles were excluded. Provide an exact number with specific exclusion criteria.
You assess articles using the SIGN tool. Provide more detailed information on the quality of the articles included.
Figure 1: Given that you did the pre-test before the intervention was initiated, this should also be evident from Figure 1. Move the pre-test box one higher.

Materials and Methods:

3.2. Sample
Add a citation for the G*Power.

3.3. Intervention

Table 1: I recommend adding intermediate lines to Table 1, as this will make it easier to see the transition between weeks and methods.

3.8. Data Analysis

Add information on what is the p-value you considered 0.05 or less?

Results:

Figure 2: If there are multiple figures, they should be listed as (a) Description of what is in the first figure; (b) Description of what is in the second figure.

Discussion:
Add research limitations related to the methodological approach.

References:
The literature is content relevant, but I recommend more up-to-date literature.
For references before the page range, add pp.

Author Response

(The authors gave the same response as above.)
